# Characterizing Highly Effective Technology and Engineering Educators

Andrew John Hughes [1,*] , Tyler S. Love [2] and Kristine Dill [3]

1   Education Leadership and Technology, California State University, San Bernardino,
    San Bernardino, CA 92407, USA
2   Department of the Built Environment, University of Maryland Eastern Shore, Baltimore, MD 21230, USA;
    tslove@umes.edu
3   Career and Technical Education, California State University, San Bernardino, San Bernardino, CA 92407, USA;
    k.dill9274@coyote.csusb.edu
*   Correspondence: andrew.hughes@csusb.edu

**Abstract:** There have been numerous definitions and models proposed in attempts to better conceptualize effective educators; however, there is no consensus on a definition or model that characterizes effective educators in all contexts. Specific to technology and engineering (T&E) education, the *Standards for Technological and Engineering Literacy* (STEL) proposed three elements (core standards and benchmarks, T&E practices, and T&E contexts) for standardization of instruction to ensure more effective T&E educators. However, this requires educators to possess a broad spectrum of integrative knowledge and practices to guide authentic T&E teaching and learning experiences, something which the literature has shown is not always correlated with teaching experience. This article examines various definitions and characteristics of effective educators as presented throughout the literature considered within the context of T&E education. The information presented in this article has implications for helping educators, educator preparation programs, and professional development providers identify and develop competencies that the literature suggests can result in more effective T&E educators.

**Keywords:** engineering education; technology and engineering teachers; STEM educator effectiveness; teacher preparation; pedagogical content knowledge (PCK); metacognitive awareness

## 1. Introduction

The purpose of this article is to help educators, educator preparation programs (EPPs), and professional development (PD) providers identify and develop the competencies of highly effective technology and engineering educators (T&EE). Historically, there have been numerous definitions of effective teaching and educators; thus, defining educator effectiveness has proven elusive [1]. The International Technology and Engineering Educators Association (ITEEA) developed (Figure 1) the *Standards for Technological and Engineering Literacy* (STEL) [2], which were developed to help K–12 educators plan and deliver effective technological and engineering literacy instruction. The inner standards octagon in Figure 1 represents the core concepts, including detailed benchmarks organized by grade band, that students should be able to apply through various T&E practices within a broad range of T&E contexts. The middle practices octagon represents T&E practices derived from 21st Century Skills and Engineering Habits Of Mind [2]. The practices reflect students' knowledge, skills, and dispositions to successfully apply the standards and benchmarks in different T&E context areas [3]. The outermost octagon represents the range of contexts in which students can potentially apply T&E concepts and practices. The STEL specifies that these eight context areas are not all-inclusive but allow teachers flexibility in addressing the standards relative to their students, region, and community [3]. Unlike the standards and benchmarks, students should not be expected to master all eight T&E context areas

in the STEL [2]. Moreover, the STEL also acknowledges that the T&E context areas may evolve as new technologies emerge. The STEL was created with those types of emerging changes in mind, allowing for T&E teaching and learning to remain relevant for students, schools, communities, and society. Brown and Antink-Meyer [4] determined that the STEL adequately represents the seven features of the Nature of Engineering Knowledge (NOEK). However, their study also reported that T&EEs had incomplete and sometimes incorrect knowledge related to each feature of the NOEK [4].

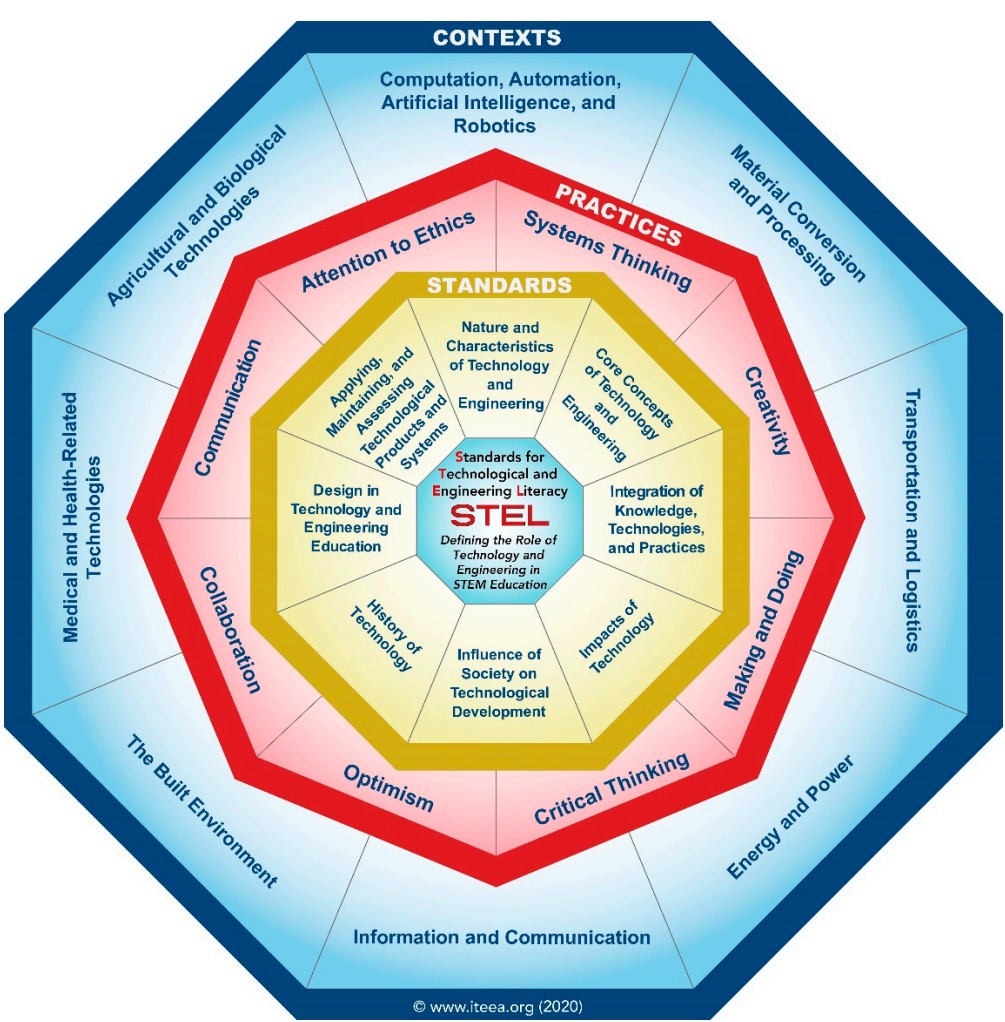

**Figure 1.** Graphic organizer of STEL. *Note:* Reprinted from *Standards for Technological and Engineering Literacy: The Role of Technology and Engineering in STEM Education* [2] p. 11, by the International Technology and Engineering Educators Association (ITEEA). Copyright 2020 by ITEEA. Reprinted with permission.

When planning instruction based on the STEL, educators should start with the standards and then consider the contexts in which the concepts may be applied. The instructional process could include interdisciplinary contexts and numerous practices and/or context areas [2,3]. Figure 1 shows that to become a highly effective technology and engineering (T&E) teacher, educators must possess a broad spectrum of knowledge and practices to help form their foundation [5–7]. Some studies have found that teaching experience alone does not result in more effective T&E instruction; however, educators' formal and informal science [8], T&E [6], and laboratory safety [9] preparation experiences were found to be significantly associated with more proficient T&E teaching. These preparation experiences help form the foundational abilities of highly effective educators, including metacognitive awareness, pedagogical content knowledge (PCK), maintaining high ex-

pectations, supporting student success, classroom management, continual improvement, and other abilities. In addition, many studies, such as that by Phillips et al. [10], discuss positive characteristics and special personal qualities that effective educators possess. For example, for students to achieve their goals, educators must develop trusting, student-centered educator–student relationships in democratic classrooms where students feel empowered [11].

Effective teaching involves building upon other foundational skills that help educators get to know their students better, something essential for student success [1]. However, considering the experiences of students is only one aspect of becoming a high-quality educator. Many other educator characteristics also play an essential role in educator quality and student perceptions of educator quality, including knowledge and experience in the discipline, educator credibility, mutually respectful relationships, positive attitude, educator clarity, and many other essential characteristics [12]. When educators exhibit positive qualities to their students, they begin to trust the educator to provide a positive learning experience [13]. The foundational characteristics of high-quality educators include having developed greater metacognitive awareness, PCK, and thoroughly utilizing critically influential teacher practices during the teaching and learning process [5,6,12]. Metacognitive awareness can be defined as "the ability to recognize and regulate one's own thinking in real time" [5] p. 25. Educators that lack metacognitive awareness will struggle to adapt to the constantly evolving educational environment and to help students develop their metacognitive awareness [5]. Further complicating this is the increased focus on integrative practices within science, technology, engineering, and mathematics (STEM) education contexts, which will require educators to develop a broad range of PCK to thoroughly engage students with rigorous instruction that coherently integrates content and practices from multiple disciplines [6,12,14]. Consequently, with the importance of the interdisciplinary technological and engineering design-based content that T&EEs address, it is essential to identify the specific characteristics that T&EEs already possess and develop other high-impact practices that help create and sustain successful learning environments.

## 1.1. Defining Effective Educators

Simply stated, effective educators lead to successful student learning outcomes [15]. Brockett [16] believed an influential educator has seven essential qualities or attributes. Brockett [16] used the acronym "TEACHER" to describe seven essential qualities that effective educators possess and should implement within the learning environment (Table 1).

**Table 1.** Seven Essential Qualities of Effective Educators [16] p. 10.

| Quality | Description |
|---|---|
| **T**rust | Creates a successful and safe learning environment where the learners feel free to explore ideas and share their views without fear of criticism from the educator or other learners. |
| **E**mpathy | Developed by listening carefully to the learners' concerns. An effective educator can understand the learner's feelings and convey this understanding to the learner. |
| **A**uthenticity | Another word for "genuineness", which means being yourself. An authentic educator does not put on a front attempting to be someone they are not. |
| **C**onfidence | Effective educators are confident. They are confident about what they know and how to share it with their learners. Confidence grows with experience, and the educator's confidence increases as the learners succeed. |
| **H**umility | An effective educator demonstrates mastery of a topic without being self-important. The educator accepts that they sometimes make mistakes and do not take themselves too seriously, which helps build trust with the learners. |

**Table 1.** *Cont.*

| Quality | Description |
|---|---|
| Enthusiasm | Whether the educator is "bubbly" or more reserved, conveying enthusiasm and passion through the love of the topic and an ability to share demonstrates excitement for students developing an understanding of content. |
| Respect | Respectful educators appreciate and value learners, even if they do not share the same beliefs or ideas. Such educators do not try to intimidate, humiliate, or use their power to persuade or threaten learners to change their beliefs. |

Brockett [16] further indicated that these seven educator characteristics are connected and balance each other out. Therefore, as an educator develops and refines these qualities and implements them in the classroom setting, it creates a positive classroom environment, leading to further teacher improvement and successful student learning and disposition development.

## 1.2. Learners' Perspective of Positive Educator Characteristics and Practices

Similar to the authors' personal experiences, decades of literature investigating the recruitment of T&E teachers suggest that students commonly have fond memories of an influential T&EE. Studies over the past half-century have consistently found T&EEs to be the most influential factor associated with one's decision to become a T&EE [17]. Whether students recognize the remarkable impact of their T&EE while still in school or after graduating, the authors have observed learners valuing the influence their T&EE had on their education. Love et al. [17] described instances where T&EEs influenced 15 and 28 students to become T&EEs. Upon closer examination of Love et al.'s [17] examples, Volk [18] opined that students in the classes of these influential T&EEs might have been drawn to the profession by engaging in industrial arts–oriented projects that were no longer a focus of national standards and T&EE preparation programs. Volk [18] believed these fun, hands-on projects were enjoyable, memorable, and attracted students to the teaching profession. The push for a more academic focus and integration of science and mathematics within T&E education may not be as attractive to students; this could be one reason for retention issues in T&EE preparation programs [18].

While not specific to T&E education, prior research offers valuable insight into the positive characteristics of students' most memorable teachers [19]. Over 15 years, Walker [19] examined information from thousands of students, including essays that allowed students to provide examples of how their teachers inspired them. Students described their most memorable teacher's unique personal qualities and characteristics in these essays. In addition, the students reflected on educators that made the most significant impact by effectively teaching the subject matter. In addition to the essays, group discussions were held to uncover what students believed constituted an excellent teacher. After comparing the information from the essays and discussions, Walker [19] reported twelve themes or characteristics of the students' favorite teacher (Table 2).

**Table 2.** Twelve Characteristics of Students' Favorite Teacher [19].

| Characteristic | Description |
|---|---|
| Preparation | Came to class prepared. |
| Attitude | Maintained positive attitudes about teaching and the students. |
| Expectations | Had high expectations for students. |
| Teaching Style | Had a creative teaching style. |
| Fairness | Treated and graded students fairly. |
| Approachability | Was approachable. |
| Sense of Belonging | Cultivated a sense of belonging in the classroom. |
| Compassion | Was compassionate. |
| Sense of Humor | Had a sense of humor and did not take everything seriously. |
| Respectful | Respected students. |

**Table 2.** *Cont.*

| Characteristic | Description |
|---|---|
| Forgiving | Was forgiving and did not hold grudges. |
| Authentic | Admitted their mistakes. |

Many items reported in Walker's [19] study align with factors that positively influence student achievement [12]. All students have unique perspectives on educator quality. Therefore, students' feedback is instrumental in determining what pupils value in their educators [12]. Educators might consider having students write brief essays focusing on the specific traits of their most memorable teacher. Having students write essays is an effective strategy to help instructors gain insight into traits valued by their students and allow educators to reflect on improving their teaching skills [12].

### 1.3. Learners' Perspective of Problematic Educator Characteristics and Practices

In addition to examining and striving to develop positive educator characteristics, T&EEs must be aware of and learn from problematic characteristics. Phillips et al. [10] indicated that educators could retain problematic negative qualities despite understanding effective qualities. For example, T&EEs must understand that adolescent students have unique experiences that can impact their learning; however, this is commonly only associated with adult students. While T&EEs' positive characteristics and practices can create a thriving learning environment, educators must refrain from practices that negatively impact learners. For example, educators perceived as not understanding, disrespectful, dismissive of learners' prior knowledge and experiences, and disregarding of learners' time resulted in disinterested and unengaged students [10]. Another negative characteristic that impacted students' learning was the perceived lack of educator credibility [10,12].

For newer and younger educators in particular, their perceived lack of credibility negatively impacted student learning [10,12]. Compounding newer and young educators' perceived lack of credibility was their anxiety and apparent lack of confidence in the learning environment [10,12]. While an educator's apparent lack of confidence was an issue, research also found that educators' arrogance and lack of respect for students negatively impacted students' learning [10]. Moreover, when students perceived educators as rigid and disorganized, the students became disengaged in learning [10]. Students commonly perceive negative educator qualities (e.g., being disorganized, disrespectful, and arrogant) as hindering their learning [10]. As one of the author's former principals frequently stated, "A student's perception is their reality".

Recognizing these negative educator qualities as a hindrance to student learning is essential for helping T&EEs develop trusting relationships with students. To further help contrast these negative perceptions, Berman [1] indicated that the learning environment should be safe, non-threatening, positive, and collaborative. These recommendations from Berman [1] reminded the authors of the saying that students must Maslow before they Bloom. This saying is in reference to Maslow's hierarchy of needs [20] and Bloom's cognitive [21], psychomotor [22], and affective taxonomies [23]. Ultimately, students' and teachers' perceptions of learning environments can vary, and each can have a different perspective [1]. Therefore, EPPs and school administrators should consider providing greater support for educators, especially newer and younger educators. Support can help counter these perceptions while developing a more positive and successful learning environment [12].

### 1.4. Cognitive Appropriateness

Another foundational skill of effective T&EEs is a thorough understanding of cognitive appropriateness. When T&EEs present too advanced or simplistic content for their students, it can lead to frustration, disengagement, and a lack of motivation to learn. Willingham [7] suggested that students avoid thinking unless the cognitive conditions are appropriate. As a result, the learner becomes disinterested and gives up. Willingham [7] highlights this point by encouraging educators to establish appropriately challenging learning to instill learners' curiosity. In training teachers, the authors use the story of Goldilocks and the Three Bears to simplify the idea of cognitive appropriateness. Utilizing stories like Goldilocks and

the Three Bears is a psychologically privileged method to help future teachers begin to conceptualize learning theories, paradigms, and concepts, including scaffolding, zone of proximal development, and constructivism [7]. In the story, Goldilocks tries the porridge until she finds the bowl at the right temperature to consume. Goldilocks finding porridge at the right temperature for her to consume exemplifies an essential characteristic of T&E education. Based on the practical, hands-on, and project-based learning that happens in T&E education, the student, with guidance from the T&EE, learns to engage with the content at their appropriate cognitive level, just like Goldilocks finds the porridge at her desired temperature.

Disciplinary concepts and core ideas commonly addressed in T&E education can also be covered in science and mathematics courses [6,24]. However, these concepts are covered using approaches common to science and mathematics with minor but significant differences in terminology, structure, and focus compared to a T&E education approach [25]. Therefore, it is of utmost importance for STEM educators to possess disciplinary awareness in the discipline they teach and collaborate with educators in complementary disciplines for a more in-depth integrative learning experience [26]. Moreover, disciplinary awareness enables educators to bridge connections between different STEM fields, illustrating the interdependence and associative nature of the STEM disciplines. This holistic approach encourages students to think critically, solve problems creatively, and apply knowledge across disciplinary boundaries, preparing them for the dynamic and interdisciplinary challenges of the future. Ultimately, possessing disciplinary awareness empowers STEM educators to cultivate a rich and comprehensive educational experience, equipping students with the skills, knowledge, and a mindset necessary for success in an increasingly interconnected world.

Although the brain is not particularly efficient at thinking, the brain enjoys successful mental activity [7]. For example, suppose an individual perceives value from engaging in a mental task. Their curiosity encourages them to act; however, curiosity is fragile [7]. To help maintain students' curiosity, the T&EE can work with students to establish operational definitions and quality standards to clearly communicate high expectations while supporting students in reaching such expectations [7,12].

As the T&EE develops a supportive relationship with students, they can engage students in increasingly demanding cognitive content by utilizing appropriate learning supports for individual students. For example, suppose a student lacks background knowledge on a specific topic. In this case, they will quickly lose interest or become easily frustrated. Ideally, before the student loses interest, the T&EE should evaluate the support the learner requires to engage with the topic successfully [5]. The T&EE could select from several strategies, including adjunct aides, class discussion, the jigsaw method, or the use of quality improvement tools based on the timeliness of their evaluation and the support required by the student [12].

It is also equally crucial for T&EEs to remember the limitations of working memory. Teaching is a cognitive skill that requires manipulating the working memory of one's brain [7]. Therefore, teaching is demanding of an educator's working memory. The factual and procedural knowledge needed for effective teaching must transition from working memory to long-term memory to help reduce cognitive demand [7]. The brain's long-term memory can store declarative, procedural, and conditional knowledge [5,7]. Cognitive psychologists refer to the brain's working memory as a site of awareness and thinking, and a mental place where several things are juggled at once [7]. The brain needs sufficient room in the working memory to think effectively; however, an individual's working memory has limited space. If there is too much information, some information will be dropped from one's working memory [7].

There is only so much information a T&EE's memory can store simultaneously. Unless the concept is simple, one's working memory can become overloaded when there are "lists of unconnected facts, chains of logic more than two or three steps long, and the application of a just-learned concept to new material" [7] p. 15. If this happens, the T&EE must

slow the pace and implement helpful memory aids, such as writing the information on a strategy evaluation matrix or regulatory checklist, so the learner can successfully manage the amount of information stored in their working memory [7,27]. In addition, the learner can use metacognitive strategies such as taking notes and saving them for future reference. For T&EEs to be highly effective, they must understand how students learn and implement metacognitive and self-regulatory practices.

*1.5. Metacognitive Awareness*

Metacognition differs from cognition. Cognitive skills are essential to performing a task, whereas metacognition is essential to understanding how, when, and why the task is performed [27]. Hughes [5] described a metacognitive awareness framework, depicted in Figure 2. Metacognition has inaccurately been commonly equated to singular processes such as reflection, which are only part of a metacognitive awareness framework [28]. Metacognitive awareness has been a research topic for nearly three decades; yet, developing T&EEs' and students' metacognitive awareness remains relevant. Researchers have documented a link between educators' metacognitive skills and the effectiveness of their teaching practices [5,29]. Effective T&EEs possess higher metacognitive awareness levels, resulting in heightened learning capabilities that help translate PD experiences into classroom improvement [5]. Similarly, metacognitive research posits that students' metacognitive awareness is crucial for improving their learning [5]. Consequently, T&EEs who lack metacognitive awareness often have difficulty adapting to constantly changing educational environments [5,28].

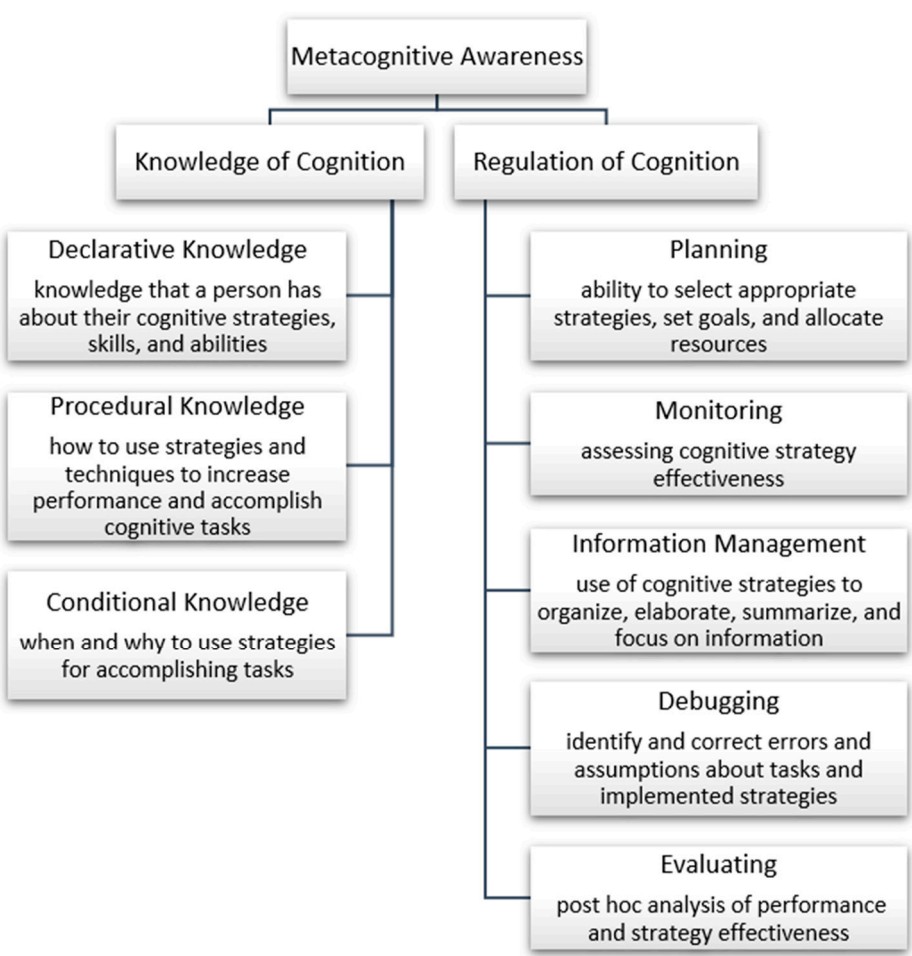

**Figure 2.** Metacognitive Awareness Framework.

Schraw [27] discussed the significance of educators modeling metacognition for their students and using tools to help students develop metacognitive awareness. Though there are many methods an educator can use to help develop students' metacognitive awareness, modeling is fundamentally important [27]. Other methods for developing students' metacognitive awareness include using tools like strategy evaluation matrices and regulatory checklists [27]. Metacognitive awareness development tools commonly focus on improving students' information organization, elaboration, and management, as well as the students' self-monitoring, evaluation, reflection, and help-seeking behaviors. Schraw [27] indicated that empirical studies using instructional tools helped improve student learning by increasing metacognitive knowledge and regulation (Figure 2). Therefore, metacognitive awareness, and the associated skill sets, are considered beneficial for educators and students. The STEL addresses the development of students' metacognitive capabilities in Standard 2: Core Concepts of Technology and Engineering and Standard 7: Design in Technology and Engineering Education [2]. However, as Hughes [5] indicated, T&EEs with a more developed metacognitive awareness will be able to better assist students with developing their metacognitive awareness.

### 1.6. Pedagogical Content Knowledge

Practices associated with an educator's metacognitive awareness are at least semantically associated with their PCK. Metacognitive awareness and PCK are similar, with subtle essential differences. While a T&EE's metacognitive awareness helps manage educational complexity, their PCK helps reduce the cognitive load on working memory during teaching. PCK is the complex combination of content and pedagogical knowledge associated with high-quality teaching [6]. An educator's PCK is a unique blend of understanding of content knowledge and ideal methods for presenting content to students [30]. An educator's PCK represents the content and pedagogical knowledge they were able to transition from working memory into long-term memory throughout their educational experiences. The STEL emphasizes that the unique PCK of T&EEs is essential for enhancing students' level of technological and engineering literacy [2]. The STEL also highlights the need for T&EEs to develop the PCK required for authentic T&E learning environments: "The technology and engineering contexts and practices provide comprehensive details about the unique pedagogies used in technology and engineering learning environments" [2] p. 6.

Love and Hughes [6] proposed that changes to T&EE preparation programs were needed to better develop educators' PCK for teaching content and practices within authentic T&E contexts. One of their recommendations was that T&EE preparation programs thoroughly focus on content knowledge development, especially in the science disciplines [6]. Rose et al. [31] also recommended that T&EE preparation programs should require higher levels of mathematics and science courses to develop better content knowledge needed to teach authentic T&E concepts. The idea that an educator's depth of content knowledge can influence their level of PCK is well documented throughout the literature [6]. Rose et al. [31] and Love and Hughes [6] highlighted the importance of T&EEs developing more profound content knowledge to improve their teaching of authentic T&E concepts [6]. This relates to Willingham's notion that "factual knowledge must precede skill" [7] p. 19, which is valid for both educators and students. T&EEs must build their content knowledge to a sufficient level for the concepts and grade level taught. Then, they can transfer that knowledge to students through continuous practice involving applied pedagogical knowledge [6,7]. At the higher education level, studies have also highlighted challenges with facilitating interdisciplinary learning opportunities due to the specialized, discipline-specific content knowledge and pedagogical practices required of instructors [14]. This further exemplifies the importance of collaboration to deliver interdisciplinary instruction in greater depth [26].

### 1.7. Continual Improvement Process

As with anything in life, teaching requires deliberate practice to improve. Willingham suggested that "teaching, like any complex cognitive skill, must be practiced to be

improved" [7] p. 147. Gaining competence and improvement are the primary reasons to practice teaching [8]. As such, becoming a highly effective educator does not happen overnight. However, an educator should not assume that practice is synonymous with experience [7]. More than just practice, developing into an effective educator takes years of concerted effort focused on continual improvement [7]. T&EEs must continually and consciously work to improve their content knowledge, develop their teaching skills and practices, seek feedback from others, undertake activities towards improvement (e.g., self-reflection, PD), and consider many other facets of teaching [7]. Willingham [7] suggested that educators focus on three steps to improve their teaching skills:

1. Consciously trying to improve
2. Seeking feedback on teaching
3. Undertaking activities for the sake of improvement

As educators focus on improving students' habits of mind, which are reflected in the T&E practices within the STEL [2], they must also focus on improving their own habits of mind. Psychologists use the expression "habits of mind" to describe aspects of intelligence. Lucas and Hanson stated, "A critical distinction between habits of mind and other popular ways of describing individual learning differences, for example, non-cognitive skills, is that habits of mind or learning dispositions, are not fixed traits" [32] p. 5. Instead, an educator's habits of mind are capable of development. The view that learning performance can improve through deliberate effort and practice is what Dweck [33] discussed with the idea of a "growth mindset" [7,32]. T&EEs who believe that their abilities can change, work hard, try different strategies when they get stuck, and see failure as an opportunity to grow can improve their mindset regarding teaching. This, in turn, can help to improve T&EE development toward becoming a more effective educator.

## 2. Discussion

Numerous recommendations were made throughout this article. It would be inappropriate for the authors to suggest that the highly effective T&EE characteristics described in this article represent an exhaustive list. Instead, these recommendations help form a foundation for effectiveness and potentially serve as a model for continual educator improvement. The recommendations in this article should be viewed systematically rather than as a means to an end concerning T&EEs' continual improvement process. An effective educator should be able to self-evaluate to determine what they are doing well from the characteristics provided and what they could work on improving [5,28]. The view that teaching performance can be improved through deliberate effort and practice is related to Dweck's [33] discussion of the growth mindset. Educators who believe that "their abilities can change, who work hard, try different strategies when they get stuck, and see failure as an opportunity to grow" likely foster similar behaviors in their students [32] p. 6. T&EEs must reflect on whether they catalyze students' interest in T&E and their success. In other words, is the T&EE serving as a gatekeeper or a gateway for student success? Effective T&EEs continue to ask and reflect on that question throughout their careers. If T&EEs find themselves answering that they are the gatekeeper, they should make an effort to institute changes that hopefully result in improvements related to the characteristics listed throughout this article. As Hughes indicated, "the complex thinking involved with the interdisciplinary approach of content and pedagogical knowledge required for engineering education requires teachers to cognitively prepare, monitor, adapt, and reflect" [28] p. 18.

## 3. Conclusions

This article provides greater clarity of the definition of an effective educator, the characteristics of memorable educators, the qualities of a highly effective educator, and educator characteristics that students have reported as a hindrance to their learning. T&EEs should start by self-evaluating and changing their most problematic or ineffective educator characteristics and qualities. As indicated, improvements will take concerted efforts and years of deliberate practice focused on developing and implementing metacognitive

awareness, PCK, and a growth mindset, among others. Furthermore, an educator getting to know their students in greater depth can help improve the learning environment and the instructor's effectiveness. Finally, an educator that takes the time to become acquainted with their students demonstrates respect, value, and fosters a positive and successful learning environment.

Continual educator improvement requires more than the critical concepts discussed in this article. Educators should not assume that there is a "one size fits all" or "best way" to improve their teaching effectiveness. It is recommended that educators experiment with implementing different research-based practices, keeping in mind from a psychological perspective that it takes roughly 21 days to replace an old habit with a new one. Furthermore, automating new habits takes roughly 66 days [7]. The authors emphasize the significance of educators taking their time to improve, because while educators are adjusting, so are their students. Finally, some of the concepts and strategies recommended for positively influencing student performance based on Hattie's [12] work will require T&EEs to develop their growth mindsets. Effective T&EEs should remain steadfast in improving their students' academic success.

**Author Contributions:** Conceptualization, A.J.H. and K.D.; writing—original draft preparation, A.J.H., T.S.L. and K.D.; writing—review and editing, A.J.H. and T.S.L. All authors have read and agreed to the published version of the manuscript.

**Funding:** This research received no external funding.

**Institutional Review Board Statement:** Not applicable.

**Informed Consent Statement:** Not applicable.

**Data Availability Statement:** No new data were created or analyzed in this study. Data sharing is not applicable to this article.

**Conflicts of Interest:** The authors declare no conflict of interest.

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
