# Peer review of "Characterizing Highly Effective Technology and Engineering Educators"

_education, doi:10.3390/educsci13060560_

Round 1

Reviewer 1 Report

Thank you for the opportunity to review your paper. I appreciate the time and effort to delineate the characteristics of effective Technology and Engineering Educators. The range of sources and references is broad, and will be useful for T & EE practitioners. The key thing that I am missing, however, is that one important characteristic for a successful educator in STEM is awareness of the nature of the knowledge itself. Learning and applying a physics-based concept in an engineering context is very different from learning and applying a technology, or for that matter, a mathematics-based concept. There is a great deal of work on these differences in the Sociology of Education. While the authors do not need to go into this, it really would be important to acknowledge the different disciplinary bases in engineering and technology education, and that these disciplinary organising principles have implications for student learning, as well as teaching. In other words, part of the Cognitive Appropriateness section could simply include this disciplinary awareness dimension.

A key literature review paper that looks at this is perhaps the following:

Winberg, C., Adendorff, H., Bozalek, V., Conana, H., Pallitt, N., Wolff, K., ... & Roxå, T. (2019). Learning to teach STEM disciplines in higher education: a critical review of the literature. Teaching in Higher Education, 24(8), 930-947.

I have two small recommendations for attention:

Lines 54 & 55: I suggest you support the following statement with a suitable reference – perhaps Brockett: “When an educator exhibits positive qualities to their students, the students subsequently trust the educator to provide a positive learning experience”. There is not necessarily a causal relationship here, but Brockett’s holistic TEACHER model suggests all the attributes would contribute to students’ trust, and therefore, positive learning outcomes.

Section 1.4 Cognitive appropriateness: The Goldilocks metaphor – while ‘sweet’ – is really Vygotsky’s Zone of Proximal Development, or a foundation principle in socioculturally mediated learning. If the focus of the journal is to represent research-informed, scholarly approaches, then I suggest the authors link the metaphor to the principle of ZPD – which differs for every student. Secondly, cognitive appropriateness in STEM really should take cognisance of the different disciplinary organising principles.

I suggest a proof reader for minor language and syntax corrections n the abstract specifically.

Author Response

Reviews and Author Responses for Manuscript: Education-2416604

Title: Characterizing Highly Effective Technology and Engineering Educators

Authors’ Overall Comments: We want to first thank the reviewers for their expertise and providing thoughtful and constructive feedback to improve the clarity of this manuscript. We have provided comments regarding how we addressed each concern below. Edits in the manuscript have also been added using red text to help reviewers see what changes we made related to the responses below.

Editor’s Comments:

In addition, please expand the word count of your article. We suggest a minimum word count of 4,000 words for the main text (excluding References and author information).

Authors’ Response: We have added additional relevant literature at the reviewers’ request. That has now increased the word count above 4,000 words.

Reviewer 1:

Thank you for the opportunity to review your paper. I appreciate the time and effort to delineate the characteristics of effective Technology and Engineering Educators. The range of sources and references is broad, and will be useful for T & EE practitioners. The key thing that I am missing, however, is that one important characteristic for a successful educator in STEM is awareness of the nature of the knowledge itself. Learning and applying a physics-based concept in an engineering context is very different from learning and applying a technology, or for that matter, a mathematics-based concept. There is a great deal of work on these differences in the Sociology of Education. While the authors do not need to go into this, it really would be important to acknowledge the different disciplinary bases in engineering and technology education, and that these disciplinary organising principles have implications for student learning, as well as teaching. In other words, part of the Cognitive Appropriateness section could simply include this disciplinary awareness dimension.

Authors’ Response: Content was added on lines 197-211 to address disciplinary awareness.

A key literature review paper that looks at this is perhaps the following:

Winberg, C., Adendorff, H., Bozalek, V., Conana, H., Pallitt, N., Wolff, K., ... & Roxå, T. (2019). Learning to teach STEM disciplines in higher education: a critical review of the literature. Teaching in Higher Education, 24(8), 930-947.

Authors’ Response: Thank you we have added this reference to the article and discussed it where applicable. One main difference was this article focused on higher education faculty and students, whereas our manuscript focused on K-12 teachers and students. We made appropriate connections with those differences in mind.

I have two small recommendations for attention:

Lines 54 & 55: I suggest you support the following statement with a suitable reference – perhaps Brockett: “When an educator exhibits positive qualities to their students, the students subsequently trust the educator to provide a positive learning experience”. There is not necessarily a causal relationship here, but Brockett’s holistic TEACHER model suggests all the attributes would contribute to students’ trust, and therefore, positive learning outcomes.

Authors’ Response: Although Brockett’s work is a good suggest for citing, we decided to reduce the causal relationship and cite Stronge’s work.

Section 1.4 Cognitive appropriateness: The Goldilocks metaphor – while ‘sweet’ – is really Vygotsky’s Zone of Proximal Development, or a foundation principle in socioculturally mediated learning. If the focus of the journal is to represent research-informed, scholarly approaches, then I suggest the authors link the metaphor to the principle of ZPD – which differs for every student. Secondly, cognitive appropriateness in STEM really should take cognisance of the different disciplinary organising principles.

Authors’ Response: There is a lot of depth to the Goldilocks story. Although, Vygotsky’s Zone of Proximal Development is one of the key concepts, there is also the cognitive benefit of storytelling, and other key concepts and paradigms addressed. Content was added to this section to address these points. The STEM cognizance was addressed based on the previous comment.

Comments on the Quality of English Language

I suggest a proof reader for minor language and syntax corrections n the abstract specifically.

Authors’ Response: We learned that there were some errors that occurred when the Microsoft Word version of the manuscript was transferred over to the MDPI template. We have read through it carefully and addressed all errors in language and syntax.

Reviewer 2 Report

1-In abstract please highlight the main findings of your research, if literature review highligh the main recommendations. 

2- Introduction check reference 5 become before 1? please check all reference, put them in order or use professionel referencing software like zotero or endnote to avoid this kind of mistakes 2.

3- line 27 what is , ITEEA? please add a table with different abbreviations and their meanings

4- or example, ITEEA developed Figure 27 to depict the Standards for Technology and Engineering Literacy (STEL) [1], please paraphrase, native speaker need to review the paper before submission of revised manuscript. 

5- references [2, 4, 6]. please avoid to use multiple references for the same sentence, cite and discuss each reference apart. 

6- same for [4, 7-8].

7- please avoid to use For example, many times in the introduction

8- Fig.1 is very interesting and need more description and analysis

9- section 1.1 Defining Effective Educators needs to be reformulated 

10- for refernces no need to put page of the textbook or the paper

11- Walker’s [17]  page 120, put reference at the end of sentence. 

12- from page 130 to 140, put all in one table and add advantages and disadvantages

13- More papers published in education and other engineering educations journals can be discussed and cited in the paper

Native speaker is recommended to read and update the language of the paper. 

Author Response

Reviews and Author Responses for Manuscript: Education-2416604

Title: Characterizing Highly Effective Technology and Engineering Educators

Authors’ Overall Comments: We want to first thank the reviewers for their expertise and providing thoughtful and constructive feedback to improve the clarity of this manuscript. We have provided comments regarding how we addressed each concern below. Edits in the manuscript have also been added using red text to help reviewers see what changes we made related to the responses below.

Editor’s Comments:

In addition, please expand the word count of your article. We suggest a minimum word count of 4,000 words for the main text (excluding References and author information).

Authors’ Response: We have added additional relevant literature at the reviewers’ request. That has now increased the word count above 4,000 words.

Reviewer 2:

1-In abstract please highlight the main findings of your research, if literature review highligh the main recommendations. 

Authors’ Response: Please see that the main recommendations are addressed in the introduction and literature review. One main finding is that the literature offers numerous definitions and models of effective teaching, but these models are not specific to the broad spectrum of educator knowledge and practice that is necessary for working towards being an effective educator. This information is explained in the abstract.

2- Introduction check reference 5 become before 1? please check all reference, put them in order or use professionel referencing software like zotero or endnote to avoid this kind of mistakes 2.

Authors’ Response: This was an error due to the numbered references not being included in the abstract. That is now fixed and reference 5 in the introduction is correct.

3- line 27 what is , ITEEA? please add a table with different abbreviations and their meanings

Authors’ Response: We have added the abbreviation for ITEEA here. We checked to make sure all abbreviations were defined upon first use.

4- or example, ITEEA developed Figure 27 to depict the Standards for Technology and Engineering Literacy (STEL) [1], please paraphrase, native speaker need to review the paper before submission of revised manuscript. 

Authors’ Response: We have spelled out ITEEA here and believe it reads very clear now.

5- references [2, 4, 6]. Please avoid to use multiple references for the same sentence, cite and discuss each reference apart. 

Authors’ Response: The author guidelines from the MDPI website state that multiple references can be used for the same sentence and they provide an example similar to how we cited in our paper - https://www.mdpi.com/journal/education/instructions#references

6- same for [4, 7-8].

Authors’ Response: Same response as above.

7- please avoid to use For example, many times in the introduction

Authors’ Response: We have edited this section to only use “For example” one time.

8- Fig.1 is very interesting and need more description and analysis

Authors’ Response: We have added more description about this graphic organizer including additional citations.

9- section 1.1 Defining Effective Educators needs to be reformulated 

Authors’ Response: Section 1.1 was reformulated to include Table 1, fix the citations as requested, and offer a clear understanding for the reader.

10- for refernces no need to put page of the textbook or the paper

Authors’ Response: Using the MDPI citation guidelines we believe book chapters like Dave (1970) require page numbers to be cited. In text citations referencing quotes require page numbers next to the numbered citation. We included these to stay in alignment with the MDPI format.

11- Walker’s [17]  page 120, put reference at the end of sentence. 

Authors’ Response: This was moved to the end of the sentence and adjusted for other instances like this in this paragraph.

12- from page 130 to 140, put all in one table and add advantages and disadvantages

Authors’ Response: We have placed the items from lines 130-140 in Table 2. We have also added Table 1 to help highlight those important characteristics for readers.

13- More papers published in education and other engineering educations journals can be discussed and cited in the paper

Authors’ Response: We have added a number of other relevant citations as seen in the reference list, including Winberg et al. 2019 per Reviewer 1’s suggestion.

Comments on the Quality of English Language

Native speaker is recommended to read and update the language of the paper. 

Authors’ Response: We learned that there were some errors that occurred when the Microsoft Word version of the manuscript was transferred over to the MDPI template. We have read through it carefully and addressed all errors in language and syntax.